# Factors influencing patients on antiretroviral therapy loss to follow up: A qualitative analysis of healthcare workers perspective

Robert Kogi[1,2]*, Theresa Krah[1], Emmanuel Asampong[2]

1 Ghana Health Service, Asunafo South District Health Directorate, Kukuom, Ghana, 2 Department of Social and Behavioural sciences, School of Public Health, College of Health Sciences, University of Ghana, Legon, Accra, Ghana

* robertkogi87@gmail.com

## Abstract

Despite expanded and successful antiretroviral therapy program coverage, a large proportion of people drop out at different stages along their treatment course. As a result, treatment gains do not reach a large proportion of these groups. It has been demonstrated that around half of the patients who test Human immunodeficiency virus (HIV) positive in Sub-Saharan Africa are lost between testing and being considered for eligibility for therapy. The purpose of this study was to determine the factors that influence patients on antiretroviral therapy who lost to follow up in HIV treatment clinics in Asunafo South District, Ahafo Region. We used phenomenological qualitative research approach in conducting this study. Purposive sampling was used to select respondents, while key informant interview was used to collect the data. The major identified challenges in carrying out follow-up visits of patients on antiretroviral therapy were wrong addresses and phone numbers of clients, coupled with poor telecommunication networks, geographical relocation of clients, poor documentation of patients' information, and non-availability of means of transport. The preferred reengagement strategies identified in this study were: supply of drugs through home visits, intensive education, engaging the services of community-based surveillance officers, enhanced regular phone calls visits, adoption and use of an integrated antiretroviral therapy clinic, intensified education on HIV, and involvement of religious leaders. In conclusion, all clinicians and stakeholders should consider the identified challenges and reengagement strategies when providing antiretroviral services.

## Introduction

Human immunodeficiency virus (HIV) remains a serious global public health issue, with an estimated 36.3 million deaths attributable to AIDS-related illnesses since the epidemic's inception [1]. Since 2016, the World Health Organization has advised that all persons living with HIV be given lifelong antiretroviral medication (ART), including children, adolescents, adults, pregnant and breastfeeding women, regardless of clinical status or CD4 cell count [2].

**Data Availability Statement:** All relevant data are within the manuscript and its Supporting Information files.

**Funding:** Funding for this project was provided by Access and Delivery Partnership (ADP), supported by the Government of Japan and led by the United Nations Development Programme, in collaboration with the World Health Organization's Special Programme for Research and Training in Tropical Diseases (TDR) and PATH (Grant number: UGSPH/CIRT-M/003).

**Competing interests:** The authors have declared that no competing interests exist.

Over the years, there has been a greater emphasis in Ghana on ways to sustain long-term antiretroviral therapy (ART) benefits and lower new HIV infection rates [3]. However, patient retention in ART programs in Sub-Saharan Africa, particularly Ghana, is often poor [4], leading to loss to follow-up (LTFU). Treatment advances of HIV do not adequately reach the necessary number of adults and children who drop out of care at various points during their treatment route, even though program coverage with ART has been improved and has been quite beneficial [5].

In rural South Africa, it was noted that duplication of work or transfer of tasks and obligations from one actor to another was an issue. This took the form of delegating responsibility to nurses who may devote more time to clinical work, allowing different actors to be unaware of each other's roles, resulting in work duplication, or in other cases, some patients not receiving all needed interventions because some staff did not receive their information on work demands [6]. It was also found in Lagos, Nigeria, that some clients gave false contact information to avoid HIV-related stigma [7].

Financial constraints, lack of access to treatment while traveling or working away from home, ART medication stock-outs or shortages at the clinic, medication side effects and misconceptions regarding ART and the idea that using complementary or alternative therapies could treat HIV infections were the most common reasons given in a qualitative study conducted in Kumasi [8]. However, it has been stated that it is mostly dependent on the professional's capacity to include and urge patients to attend future appointments and collaborating with them to address the causes of their potential disengagement [9]. Strength-and motivation-based approaches, such as learning about the patient's needs and working with them to identify the best solutions for addressing each unique engagement obstacle, may also be very helpful. Furthermore, there is a fundamental set of LTFU requirements that can guarantee reengagement. These include ranking urgent requirements first, categorizing needs into short- and long-term categories, depending on partner organizations to supply support services, and being given priority enrolment in programs like addiction treatment centres or social work help [9].

The antiretroviral coverage in Ghana is estimated to be at 45% among PLHIV [10]. In the Asunafo South District of Ahafo Region, report on the District Health Management System (DHIMS2) for 2020 and 2021 showed that out of 3,090 and 4,017 HIV patients who were on ART, some 1,593 and 1,564 patients have respectively stopped treatment due to loss to follow up [11]. Consequently, LTFU raises HIV patients' hospitalizations, morbidity, and mortality from AIDS-related causes and adversely affects the immunological benefits of ART [12]. To the best of our knowledge, there was no study conducted so far in the study area to explore the specific reasons for HIV patients' loss to follow-up from ART. It's crucial to understand how and why people discontinue ART treatment since keeping patients on ART and guaranteeing adherence to treatment are important factors in determining positive long-term results. We therefore conducted this study to explore the factors influencing patients on ART loss to follow up and to identify the reengagement support they need to remain on the treatment.

## Materials and method

### Study design

This study employed phenomenological qualitative research approach to allow for more understanding on the causes of loss to follow-up and retention on the ART program in the Asunafo South District.

## Study setting

The study was conducted in the Asunafo South District in Ahafo Region of Ghana. The Asunafo South District is one of the six (6) administrative districts in the Ahafo Region of Ghana. According to the 2020 population and housing census, the district had a total population of 93,619 with a total number of children 0–11 months and women in reproductive age being 3,668 and 22,006 respectively [13].

In terms of health service delivery, Asunafo South District is divided into four (4) sub-districts; Kukuom, Sankore, Kwapong and Abuom with twenty (20) health facilities of which twelve (12) are functional CHPS, distributed throughout the entire sub-districts. The district has two HIV clinics rendering ART services (Asunafo South District Hospital and Sankore Health Centre). The HIV clinics run five days a week and by the end of 2021, about 4,017 clients were active in care on ART. The clinics serves patients of all characteristics and from various communities in and outside the district.

## Study population

The study population included health staff at ART clinics and facilities In-charges. Inclusion criteria were being at least 3 years at the ART sites and provides HIV care services, and manager or conduct contact tracing.

## Sampling method and sample size

The total sample used in this study was seven (7). This included one doctor, one physician assistant, three ART clinic staff, two community health workers (who were experienced in seeking patients in the community).

Purposive sampling was used to identify the health workers based on their knowledge and role in management of HIV patients.

## Study procedure

Five (5) data collection assistants with prior experience in collecting research data were trained and recruited from 25th August to 26th August 2022, on the study implementation. A written informed consent was obtained from all the participants and were asked to sign two copies (one for each of the participant and one for the research team). Key informant interviews (KIIs) were used to explore the reasons for LTFU from ART. LTFU in this study was defined as patient (s) not being on ART for more than one month [14]. It is unknown if these patients passed away, moved their care elsewhere, or had LTFU for other reasons. Distance from house to health facility, economic reasons, capacity of ART clinics, stigma, spiritual and cultural beliefs, support needs of patients, and clinic staff behaviours were all collected.

The interviews lasted between 30 minutes and an hour. Saturation was achieved after seven KIIs were conducted among different categories of health staff. All interviews were performed with the help of an interview guide created specifically for this study and were recorded with a digital audio recorder.

## Data processing and analysis

Key informant interviews were conducted in English before being transcribed directly into text form.

We used both deductive and inductive coding approaches in this study. Deductive coding was done with predefined codes or categories being applied based on existing theory or literature (in the basic and organizing themes), while elements of inductive coding was done with

**Table 1. From codes to organizing and global themes.**

| Codes | Basic themes | Organizing themes | Global Theme |
|---|---|---|---|
| • Traceable addresses<br>• Relocation of clients<br>• Change of trade<br>• Poor documentation<br>• Means of transportation | Challenges in carrying out defaulter tracing | Irregular refills | ART lost to follow up |
| • Spiritual and religious beliefs<br>• Non-acceptance of the sickness<br>• Stigma<br>• Distance and Economic issues<br>• Fear of perceived side effect<br>• Communication and Counselling<br>• Young and healthy<br>• Erratic supply of ARVs<br>• Convenience and Capacity of ART clinics | Reasons for missed appointment | | |
| • Engaging other health staff<br>• Education and openness<br>• Patient support system<br>• Complete and accurate address capturing<br>• Regular follow ups<br>• Integrated ART clinics<br>• Involvement of religious leaders<br>• Discouraging ART clinic days | Preferred re-engagement | Retain on ART | |

themes and patterns allowed to emerge directly from the data (labelled codes) in Table 1. The data was organized in Nvivo 11 to generate the themes, and then thematic analysis was performed.

## Ethical consideration

Ethical approval was sought from the Ghana Health Service Ethics Review Committee (GHS-ERC), with approval number: **GHS/ERC/021/06/22**. Letter of introduction were also obtained from the Ahafo Regional Health Directorate and sent to the Asunafo South District Health Directorate, which was used to seek consent from the facilities and HIV clinics. A written informed consent form was given to each participant to read before they consented to participate, by signing two copies of the form (one for each participant and one for the research team). Participants were assured of their confidentiality and anonymity of the information they provided.

## Results, discussion, conclusion

### Results

From Table 2, a greater number of respondents were 20–39 years. All the respondents were males, and a little below three-quarters (71.4%) had tertiary education and were health staff.

### Challenges in conducting patients' loss to follow-up exercise

This study found that wrong traceable addresses and telephone numbers were some of the major challenges health service providers faced in conducting loss to follow-up. In the key informant interview, it was revealed that most patients (clients) always give wrong traceable addresses and telephone numbers after they are diagnosed of the disease (HIV). It was also revealed that some clients do change their telephone numbers provided to the ART nurses during registration without they (ART nurses) being of the known. This was coupled with

**Table 2. Sociodemographic and occupational characteristics of the healthcare workers interviewed.**

| Characteristics | Frequency (N = 7) | Percentage (%) |
|---|---|---|
| **Age** | | |
| 20–39 | 5 | 71.4 |
| 40–59 | 2 | 28.6 |
| **Educational Level** | | |
| Tertiary | 5 | 71.4 |
| Other | 2 | 28.6 |
| **Gender** | | |
| Male | 7 | 100.0 |
| Female | 0 | 0.0 |
| Category/Occupation | | |
| Health staff | 5 | 71.4 |
| Community volunteer | 2 | 28.6 |

poor telecommunication network in most of the communities where the clients resided. This was emphasized that some clients do not give their specific community of residence. A few affirmations to support these are as follows:

"*Some clients will provide wrong traceable address and telephone numbers to healthcare provider. And once they do that, the person will get home and we try to call the number, it will be unreachable.*" (HW_ARTN2).

"*Basically, we depend on mobile phones, the contacts to call them. . . But this our terrain, it's always difficult for me to get them. Phone calls was the easiest, but most numbers are not going through because of the network challenges. . .*" (HW_ARTN3).

It has been found that the geographical relocation of clients from one community to another was affecting effective follow-up visits. It was also found that the change of commercial trading and location of trading among patients who enrolled on the program has the potential of influencing patients' adherence to treatment at the various ART sites.

"*. . .the person comes [and say] I stay at Noberkaw, . . . then the next day you go there, they will say he has relocated from that place. . ..*" (HW_ ARTN1).

"*So, you see the surge on Fridays when they come for their market . . .. . . .So, if those people change their trade or some other things, or their commercial activity from Kukuom, then it becomes difficult to locate them.*" (HW_ IC2)

Inaccurate capture of patients' information during entry and registration among some health workers was also found to affect contact tracing when the need arises. Also, non-availability of motorbikes to aid defaulter tracing was found to be affecting contact tracing. This has compelled ART nurses to basically depend on phone calls to do defaulter tracing. This was narrated below.

"*. . .At first is we only take the name then the date the client was tested, without any traceable address. . .. So once, the client does not come for the revisit, it ends there because you cannot track the client anymore.*" (HW_ ARTN2)

"*. . .we don't have motorbike to go to other far places.*" (HW_ ARTN1)

**Reasons for missed appointments for ART among HIV patients.**   The findings suggest that some HIV patients rely on spiritual and religious beliefs as a cause of HIV and its cure. They believe that HIV is a spiritual ailment that can be cured through religious faith and other spiritual means. This belief is further reinforced by some pastors who misinform their members that HIV is "bought" for them as a form of punishment from other people they have problem within their communities. This misinformation is causing some patients to abandon their medication, leading to an increase in defaulting rates.

This is revealed in some excerpts as captured below:

". . .*most clients believe in that religious believe. . .. Sometimes too when you call them, they'll say, I went to my pastors' church and they say this and that and that and that. So, for that matter, . . . they can't take the ARVs anymore. So religious belief is also affecting the defaulting rate.*" (HW_ARTN1)

". . .*And one major challenge too is, religious leaders inciting the populace. Because I think today, we decided to call some clients who are defaulters. . . .we did that and one man told me that, um, he went to one pastor and [he s*aid *he] has been given "Yezu Mogya" (Jesus' blood). So, once he thinks he use "Yezu Mogya", all the viruses are gone. So, he will no longer come for revisit or come for the ARVs anymore. . . .It's really affecting my defaulting rate.*" (HW_ARTN2)

"*Because they (pastors) always tell them that, "yadeʒ wei deʒ yatɔ ama wo" (That means someone bought the sicknesses or maybe the infection for them). Even today we had the experience,* [where] *a man just came in and we were attending to him. . . .and this man was saying, "Yadeʒ wei deʒ asaase ho nti na ɔbi atɔ ama ooo. . ." (That means, they are having land issue and the other person bought this kind of sickness or infection for him).*" (HW_ARTN3)

It was further revealed that during the treatment initiation for clients, some contemplate as to whether the condition that they are having is something that is spiritual, that somebody is trying to inflict on them in the spiritual realms. Associating the disease to spiritual cause and being pre-occupied in the mind that its treatment should be done spiritually was reported to be affecting clients' readiness and adherence to the ARVs medications. This was attributed to the different thoughts they received from people when they get back after the counselling. The findings in this study also showed that some herbalist influence and give assurance to patients that they can cure them from the disease using their concoctions. A few of the narration in this regard are captured below.

"*They* [clients] *seem so spiritual. . . .At first, they will still be battling in their mind as to whether the condition that they are having is something that has to do with the body in physical and borders on his or her health or it is something that is spiritual that somebody is trying to inflict on him or her in the spiritual realms.*" (HW_ IC1)

"*Those [clients] who default treatment sometimes they visit other places, including herbal people and religious bodies or herbalists, traditionalist, and they are given some concoction and tell them that they've cure them of the disease.*" (HW_ ART3)

Moreover, it has been found that some clients do not accept that they have the disease after their status is being disclosed to them. Such clients are noted to be more likely to default from the medication. Some clients also have the perception that HIV does not exist and once one is

diagnosed of it, it is the health workers who deliberately try to declare them having the disease, to use the numbers to go and get some money from the program (HIV). This was reported as:

"*People still think that the condition that you are talking about, they don't think it is in existent. Maybe you are just trying to tell them something or maybe there is some program that you are doing maybe money has come from somewhere and you people want to get some of the money.*" (HW_ IC1)

The study further found that stigma towards patients with the disease has led to a high rate of missed appointments in the clinics. Patients were afraid to attend appointments as they felt their status will be questioned. Self-stigma was also reported to affect patients psychologically. Stigma also affected the willingness of patients to receive home check-ups from healthcare workers. Moreover, the stigma associated with HIV has resulted in difficulty in getting patients to agree for health workers to check on them at home even if they are at the point of defaulting. It was found that being diagnosed of the disease has made people to chastise some patients in the community and this made them to wonder how they contracted it. A few excerpts reported in the KII and IDI interviews are captured below.

"*. . . Once the person is having the disease. . .they feel shy to come for refill.. . . Because most client even take their . . .hypertensive drugs at the OPD. So why are they taking only their drugs at one corner? So, stigma [is] also affecting the defaulting rate.*" (HW_ ARTN2)

"*Now the challenge is that because of the societal stigma associated with the condition, people do not want to be relatable. So that you can be able to know that he stays behind so and so landmark in times of following up.*" (HW_ IC2)

Another major barrier to treatment adherence is the distance between patients' homes and healthcare facilities. Some patients have to travel long distances for refill appointments, which was described as challenging due to economic reasons, such as poverty or transportation costs. Patients have reported missing appointments due to lack of funds. Some patients have to rely on healthcare workers to send drugs to them, which can also be difficult due to the long distances involved. This was narrated as below:

"*And poverty, poverty. . . . this area majority of them their work is farming. . . . and other places when you call them to come for refill it becomes a challenge. Yeah, they don't get money to come.*" (HW_ ARTN2)

This study also found that some clients fear of the perceived side effects of ARVs, specifically diarrhoea. These clients may stop taking their medication and not adhere to the treatment plan. Healthcare workers are aware of this issue and tried to encourage clients to continue taking their medication despite the side effects. Despite counselling about the possible side effects, some clients still struggle with adhering to the medication plan due to fear of side effects. This was stated as follows:

"*. . .once they take the medication home, they will say that they're having diarrhoea. I want to stop, but we keep on convincing them. . ..*" (HW_ ARTN2)

"*. . .And in addition to the challenges, um, some patients who also want to avoid the side effects of the ARVs.*" (HW_ARTN5)

According to the findings of this study, the behaviour of clinic staff towards patients is crucial in ensuring adherence to ART treatment. However, it was reported that some staff exhibited unprofessional behaviour which had negative impacts on patients. One KI narrated that some nurses demanded money from them before dispensing ARV medication, resulting in their avoidance to go to the clinic. This was reported as:

"...when I took over the ART, one guy [client] told me some nurses were demanding some money from them, before they will be dispensed ARV. ... she delivered at the maternity ward. And then, her husband came around to see her in the [ward]. She was HIV positive? So, she was discharged and then the nurse told the husband that his wife is having this disease, so they should bring her one crate of eggs. And then that led to her lost to follow up because she decided not to come again because she can't be coming here and then paying money or people will be taking some gifts from them." (HW_ ARTN2)

The situation of personalizing the disease by health workers especially during counselling was reported to be discouraging clients to adhere to treatment. The ordeal of another respondent was that there is a perception that once partners are diagnosed of the disease, they cannot give birth. These were narrated as below:

"...Now this person is not sick, so if put at the back of their mind that "wo yadeɛ, wo yadeɛ" (this your sickness, this your sickness). I mean, ahh! If the person goes back home, they will sit and say ahh! But I'm not sick. But you are telling me "wo yadeɛ, wo yadeɛ" (this your sickness, this your sickness). Since am not sick, there is no need for me to take this medication or come for the medications." (HW_ARTN3)

"...I don't know where I got that sickness from. It was my wife who first came with the information and I was later asked to come for testing and realized I also have the disease. We don't have children, my wife and I, so how do we conceive in our situation?" (HW_ARTN2)

The findings further indicated that some patients felt that they are too young to get HIV, while others also felt that they are energetic and healthy enough after they are tested positive of the disease. This makes such people later try to seek for some "interventions" elsewhere and not likely to remain on the drugs, even if they agreed to be enrolled on it. This was re-echoed that such people get back into the community and find themselves "good", even though they have started with the treatment. This was mentioned as:

"...some people too, feel that you are too young to have such a condition and at the time that you screen and find them to be positive or reactive to the test, they look at themselves and then they see that I am energetic, I look healthy, nothing shows that I have anything of that sort. They may want to go and try certain things, some sort of interventions." (HW_IC1)

"...although the person has started treatment, but they get back into the community and they look good. ...so, it's always difficult for them to come back after all, I am fine." (HW_ARTN2)

The findings of this study also revealed that the erratic supply of drugs also results in patients lost to follow up, especially patients who were orphaned children who lived with other relatives of old age. The capacity to retain patients in care and the structural conduciveness in the clinics were also reported to be limited. This mainly affected the patient's confidentiality and accessibility during clinics hours. This was disclosed below:

"*. . .. Some of this people too before you know, they are living with a certain grandmother who is also finding it tough to take care of that particular child and the person checking some distances to other ART centres not getting that particular dose for the grandchild, will also tend to relax. . . .. So that is another factor that make people get lost to follow up.*" (HW_IC1)

"*The place where the ART clinic is in itself is not conducive structurally. . . .So, that limit the accessibility to care. . . .. It is also a shared space with mental health unit. . . . And within the same complex as maternity. So, these three major structural situational characteristics affects accessibility. And then also it affects the patient's confidentiality.*" (HW_IC2)

**Preferred reengagement activities needed to enhance retention of HIV patients on ART.** It was disclosed in this study that to maintain and reduce clients lost to follow up, the ART nurses have made arrangement with other community health nurses to always send the drugs to clients who may agree to this arrangement. However, this strategy has also had its challenge as some of the clients do not agree to this agreement or the nurses do not show the needed commitment. This was narrated as follows:

"*. . . so we have interacted with the CHNs. So, when they go for their home visits and then those clients that we are able to reach them, we just inform the client that we have given the medication to one of our nurses to supply them. Though there are challenges but we still on course*". (HW_ARTN2)

"*. . .Most of the time we deal with community health nurses. As they're going out to the communities, the person gave us the address. So, if they go for their cwc, at least you can check on this person and let's see.*" (HW_ARTN3)

"*. . .But sometimes they will accept and then the following day, they will reject the supply or medications*". (HW_ARTN2)

Another reengagement strategy being employed by the nurses of the clinics is education on the testing of the disease and free drugs, coupled with phone calls on clients to remind them of their refill dates and encouragement to come back to care. The clients are also educated that if they don't get on with the drugs, they can still be infecting some people with the virus. It was further reported that clients are educated to take the drugs to help in suppressing their viral loads. This was disclosed as:

"*. . .we keep on interacting with them. Then we tell them everything is free from testing to medication everything is free. . . .we have been able to bring more client back to treatments.. . . because when you put the client on treatment, it will be difficult for him to infect other people. So, the client will be viral suppress, while on medication and then it won't be so much easy to infect other people*". (HW_ARTN2)

The findings also showed that the multiple months dispensing of the ARVs is one of the strategies which is helping to keep clients on the drugs. This strategy was enhanced through the support and commitment of the ART nurses received from their community-based Surveillance Officers (CSOs) with the consent of the clients. This was narrated as:

"*. . .one client is in a town, in the Bibiani road. So, she told us she can't always be coming here for the ARVs, so we gave her six months. And then we inform our CSOs and they came here and told us that if the client needs help in terms of transporting the ARVs. . ., we should*

*inform them. . . .. So, they took the client number, but we sort permission from the client first. Maybe if the client is due for viral load, they will pay the grant to come take the sample. Then if they also due for revisit, they will pay for the client to come."* (HW_ARTN2)

It was also found that during counselling, it is important to ensure that the client (s) understands the condition and help do away with any possible believe associated with the disease. This will likely enhance and improve patients' commitment and willingness to remain on the drugs. The findings also indicated that clinicians and the ART nurses should always ensure that clients take up some level of commitment or responsibility on their path to enrolment to care. This helps them imbibe some duty on their health and try to remain on the drugs. This was recounted as follows:

*"If the person understands the condition, how the nature of the management is going to be. Then the person will be committed. And if there is any loss of contact, I mean mobile contact, if the person is relocating from wherever the person stays, that spirit of commitment alone will make the person come back to give you the new address and phone number. . . .."* (HW_IC1)

*"In the counselling session too, we want the patient to also have some level of commitment or responsibility on their path to enrolment to care. But when we also let them know that they also have a role to play. In terms of taking their medication. . . . If they're having any adverse effect or event they will speak out."* (HW_IC2)

It was further noted that clients who default and are brought back must be taking through proper re-enrolment screening to find out their reasons for discontinuation of care. This helps health staff to effectively offer such clients the needed counselling and given special attention if the need be. This was reported as:

*"We must get to know from the client what caused him or her to take that decision of staying away from treatment or not coming in on appointment times to get refill and other conversation that will help us know. If the person is sincere to tell us the reason, then we'll be able to give appropriate counselling to the patient and encourage the patient if there are any other myths that must be able to be corrected in the person's mind."* (HW_IC1)

It was proposed that health workers should not let it look like the client is sick, but rather considerate on the HIV stage of the virus. This could help clients take it upon themselves and adhere to the treatment to avoid getting to the AIDS stage of the infection. It was found that there is the need for health workers to offer HIV clients, a more detailed counselling before and after they are enrolled into care. These were narrated as follows:

*". . .Maybe we should do away with the sicknesses itself and tell the client that you are have the virus now. You are not sick; you are not at the AIDS stage. So, whatever we are doing now, it's about you not getting to the AIDS stage. . . . So, we just need to make them understand that this is two different things. So, all these things you're taking it's not that you are sick, but we don't want the virus to make you sick."* (HW_ARTN3)

*"I think some of the, the counselling needs to be quite detailed enough, especially people who default treatment, and we are able to find."* (HW_IC2).

The findings also revealed that patients have the believe that going to the health facility can easily result in the exposure of their status. This is being avoided through securing patients' information using lockers instead of the previously used shelves, which are being secured from

access by other people. The patients also need to be regularly contacted via phone calls, especially those who are getting to the time of refill. During this contact, they are asked if they have experienced in side effects while taking the drugs. This will also offer the health staff another opportunity to counsel clients appropriately. This was made known that:

"*We [are] working hard to secure our folders in terms of the people living with HIV. . .. And then also doing more training for the staff in terms of patient confidentiality and then minimizing or preventing stigma associated persons leaving with HIV. Now we have secured funding that went into purposely for contact tracing. . . .we provided credit and data to the ART team. And now they're able to call the people frequently to find out how they're doing and how they're taking their medication and whether they have side effects or other things too.*" (HW_IC2)

It has also been found that the involvement of religious leaders could serve as a retention measure of people on ARV to adhere to the drugs intake because people tend to believe in them and take their words more than the health workers. It was further added that if religious leaders can understand that HIV is not a spiritual disease, and educate their members, such clients would adhere to the treatment. This was narrated that.

"*. . .. if the religious leaders understand it and talk to them, I think they will understand more than we the health workers. Some people have in mind that . . . we health workers are paid to do this work*". (HW_ARTN3)

"*. . . if these four people [Religious leaders] are able to tell them, oh, "weiɛ nyɛ homhom a deɛ*" (this is not spiritual thing). . . .So, continue taking your medication. They'll take it from there, because they believe them spiritually than us.*" (*HW_ARTN3*)

It is very important that clients who are sick from other diseases and cannot get to the clinic for ARVs continue to receive and take their drugs to avoid the worsening of their condition. The findings from this study revealed that such clients are given the opportunity to get a trusted relative or individual from their community to come for the drugs on their behalf. This was disclosed as

"*. . . when we started with them, they were able to come in for the drugs. But later they were taken down by other sicknesses. . . . So, with that, we asked for a relative they've trust. So, one I know from maybe a community B like this always let the child come for it.*" (HW_ARTN3)

The use of some designated days for clients to come to the clinics for their ARVs is also discouraged because it was reported to easily serve as avenue for people to know that one is having the disease. As a result, any day that the client is due for the drugs he or she could go to the centre for them. Clients are also always advised not to wait for their drugs to get finished before they come for refill. This is to avoid situation where they may be unable to come on their exact refill day. This was reported as:

"*. . .we don't do clinic days. . . .You know the setting we are, if you do that, they will not even come. We also advise them that when we give you let's six months, don't wait for it to get finished. Even when you are left with the last month, you can come. Ahaa, that is the way we're running here.*" (HW_ARTN3).

## Discussion

The purpose of this study was to explore the factors affecting patients' loss to follow-up in public health facilities in the Asunafo South District, Ghana.

This current study found that some patients moved long distances from their homes to the health facility (ART clinics) for their refill. This proximity can undermine tracing and make it not feasible for clinics staff. Patient who travelled far distances to the ART clinics for drugs might be because they are trying to continue their ART regimens in order to remain healthy for themselves and their families [15] and to avoid the risk of disclosure of their status in the community they live. However, on some occasions, clients faced financial challenges as noted in the key informant interview, which made them missed their clinic appointments. A similar finding was reported in Kumasi that patients had financial constraints, lack of access to treatment while traveling or working away from home [8]. A similar finding was also reported in Uganda that cost of transportation to the health facilities influenced LTFU among mothers [16].

It is unfortunate to find that some clients in this study have had negative experiences with clinic staff, as this can impact their willingness to return for healthcare services. Demanding items such as money or eggs from clients after rendering service was also reported mong the study respondents. Such behaviours can erode trust between healthcare providers and patients and may even deter patients from seeking care in the future. Similarly, reacting in a manner that hurts patients is unacceptable and can result in a breakdown of the patient-provider relationship. Empathy, respect, and clear communication are key components of patient-centred care and can go a long way in ensuring patients feel valued and supported.

To enhance patient outcomes and lower the burden of disease, it is crucial to motivate those who are lost to follow-up. By providing a welcoming and supportive environment, healthcare professionals can encourage LTFU patients to return for care and help ensure they receive the services they need to stay healthy. As a result, professionals should be trained to improve their ability to meet the needs of patients, to listen and collaborate to discover solutions, and to overcome the hurdles that may keep them out of care [17]. Other studies found that having a great relationship with clinic staff was crucial in ensuring patients stayed at the clinic [18,19].This current study's findings also supported what was reported in southern Mozambique, that one of the most frequently reported barriers to patients lost to follow up was the fear of being mistreated by health personnel [20].

Moreover, it was found that the situation of personalizing HIV by health workers especially during counselling also discouraged clients to remain and adhere to treatment. Making statements like "this your sickness" to clients during counselling made them felt worried and victimized. Hence, this practice should be discouraged among health workers. Furthermore, this study found that the inability of health workers to explain into details what HIV is and what it entails to be enrolled on the drugs affects adherence on the drugs. The capacity to retain patients in care and the structural conduciveness in the clinics were reported to be very limited in this study. This mainly affected patients' confidentiality and accessibility during clinics hours. In another study conducted in South Africa, a similar concern was reported that regarding ARV collection methods, the two patients who were interviewed had different preferences. One wanted to obtain hers from the ARV clinic's pharmacy, while the other wanted to obtain hers from an integrated drugstore so that others wouldn't find out about her status [21].

Some clients also felt that they were energetic and healthy enough after they were diagnosed of the disease. This finding corroborated with what was found in southern Mozambique, that one of the most frequently reported barriers to patients lost to follow up was the perception of being in good health [20].

Moreover, we found in this study that patients believed in their pastors to determine the cause of HIV infection and cure using religious faith and other spiritual means. We found that some clients claimed they went to their pastors, and they gave them "Jesus' blood" in their attempt to seek cure for the disease. A similar finding was reported that patients believed that God can 'cure HIV'[22]. Unfortunately, these pastors misinformed them that HIV virus is "bought" for them as a form of punishment from other people they have problems like land issues within their communities. It has therefore been found that associating HIV to spiritual cause and being pre-occupied in the minds of patients that HIV treatment should be done spiritually. This could be affecting clients' readiness and adherence to the ARVs medications. It was further found that some herbalist influenced and gave assurances to patients that they can cure them from the disease by using their herbal concoctions. Similar finding was reported in Kumasi, that their study respondents believed that HIV infections could cured by using alternative or herbal remedies [8].

It was also revealed that some clients did not believe HIV exist and did not accept that they have the disease after their status was disclosed to them. The results of this study corroborated with what was reported from the central and western districts of Lagos, Nigeria, where medical professionals faced the problem of patients providing false contact information out of fear of stigma associated with HIV/AIDS, as well as patients' refusal to accept HIV-positive results and to seek care [7]. In addition, some clients in this study also had the perception that the health workers deliberately tried to declare them with the disease, so that they could use the numbers to go and get some money from a program. Such clients are most likely to default from the medications.

Stigma and fear of disclosure of HIV status are common barriers to patient lost to follow up. As a result, stigma remained a significant problem in the treatment and management of patients in this area. According to the findings of certain key informants, stigma may have led to a number of patients missing clinic appointments. This arose mostly because of the isolated nature of clinics. This was attributed to the fact that the clients suspected that their status would be questioned, as all clients coming to the facilities take their drugs at the dispensary, while only they are to take theirs at a separate place. Moreover, the stigma associated with the disease has resulted in difficulty in getting some patients to agree for health workers to visit them at home, even if they are at the point of defaulting.

The experience of stigma among some respondents in the in-depth interview has revealed that people have chastised them in the community, and they have been wondering how they contracted it. This experience has so far affected some clients psychologically to the extent that they described it as "draining" their thoughts. This finding should alarm all stakeholders since prolonged HIV stigma prevents patients from starting ART and taking daily medicine, requires the patient to accept his/her status, and raises the likelihood of disclosure to others [23]. Similar findings were seen in Uganda, where LTFU was linked to stigma and discrimination associated to HIV, as well as insufficient support for peer educators and mothers [16].

This study also found that some clients tried to stop the intake of their ARVs because they experience some side effects like diarrhoea after taking the drugs. As a result, they put aside these drugs and did not adhere to the treatment plan. This corroborates the information that appetite loss, diarrhoea, exhaustion, and mood swings are possible adverse effects of antiretroviral HIV medications [24]. It is important to remember that failing to follow an ARV treatment plan might make the virus resistant to medications and more difficult for individuals to treat.

We also found that in situations where patients agreed for other health staff to bring the drugs to them during their home visits, some staff did not show the commitment in taking along with them the medication to the patients, whilst in some instances, the patients too did

not agree with such arrangement to be done. This calls for the designation and support of a staff member to do active HIV patients follow-up.

The findings from this study also revealed that the erratic supply of drugs also resulted in patients lost to follow up, especially patients who were orphaned children who lived with other relatives of old age.

## Challenges facing conduct of patient loss to follow-up exercise

We found in this study that most patients gave wrong traceable addresses and or wrong phone contacts after they are diagnosed of the disease. This was observed in situations where clients did not give their specific community of residence, but rather the nearest biggest community closer to them. Other clients also changed their telephone numbers provided to the ART nurses during registration without the ART nurses being of the known. This was worsened by the poor telecommunication network in most of the communities where the clients reside. This challenge was being tackled by the various clinics through instance verification of telephone numbers in the present of the clients and confirmation of their details from their various first point of entries.

This study also found that geographical relocation of patients from one community to another is affecting effective follow up visits. The change of commercial trading and location of trading among patients who enrolled on the program was reported to be affecting patients' adherence to treatment at the various ART sites. Moreover, inaccurate capture of patients' demographic information during entry and registration among health staff was found to have a direct influence on patients follow up visits when the need arises. Resources for managing HIV clients like motorbikes, communication gadgets like mobile phones were insufficient, and insufficient funding support to aid in defaulter tracing of patients.

It was revealed that other responsibilities of the profession sometimes demand a lot of times or takes a lot of time from the ART staff. This makes it difficult to give maximum attention of care to people living with HIV. This was attributed limited human resources in the healthcare sector.

It was further admitted that trainings and other activities help to upgrade the knowledge and skills of staff responsible for managing HIV and its related activities. This result partially supported the claim that giving health professionals evidence-based training and developing their communication skills will increase the likelihood of reengagement while lowering the risk of stigma, time constraints, and other issues [25,26]. It was however, found that such trainings also do interfere with the operation of the clinics.

## Preferred re-engagement activities

It was found in this present study that the ART nurses have used home visits by other community health nurses to always send the drugs to clients who may agree to this arrangement. Though this strategy has been a challenge because of improper address of the client (s) and some clients not agreeing to receive the drugs through a third person, it has been found to be a better option in keeping patients on the drugs. Another reengagement strategy needed to be employed was an exhaustive counselling on testing of the disease, drugs, and its possible side effect, coupled with phone calls on clients to remind them of the refill dates and encourage to come back to care. Re-engaging patients mostly rely on the health provider's capacity to engage and inspire patients to schedule follow-up visits and collaborate with them to address the unique issues that have caused or may cause them to disengage.

Clients who stopped coming to the clinics because they found it difficult to get money for their transportation were helped by the ART nurses to obtain the drugs. The findings also

showed that the multiple months dispensing of the ARVs is one of the strategies which was helping to keep clients on the drugs. This strategy was enhanced through the support and commitment in terms of transportation the ART nurses received from their community-based Surveillance Officers with the consent of the clients. There is therefore the need to gear all efforts towards the enhancement of community outreach services on HIV patients, by including community health workers, keeping meticulous records of every patient after ART clinic, and most likely growing the peer outreach program for health staff. This finding concurred with what was reported in Malawi that respondents often received encouragement and transportation resources from friends and family members to help them return to care [15].

This study also revealed that counselling of clients should always be targeted at letting the client (s) understands the condition and help do away with any possible believe associated with the disease. This was anticipated to enhance and improve patients' commitment and willingness to remain on the drugs. This finding supported the opinion expressed that certain patients would want additional information regarding medicine and treatment alternatives to get it, and that other patients would require encouragement to engage in active self-management and self-determination [9].

It is further found that clients who default and are brought back to care, have to be taken through proper re-enrolment screening to find out their reason (s) for discontinuation of care. This would help the health staff to effectively offer such clients the needed counselling and give them a special attention if the need be. It was further admitted in this study that encouraging and keeping patients on care is more demanding, and all health staff have to ensure that an immediate follow up is conducted when the appointment times of clients are due, and the patient is not showing up. The results of this study corroborate the claim that patients who are lost to follow-ups are more likely to be successfully reengaged in HIV care if a provider can effectively engage them, encourage them to show up for next appointments, and collaborate with them to address any causes for potential disengagement [9].

In addition, the support clients received from family members could serve as a way of encouragement to them. This our study's finding corroborated with the claim that assistance from friends and family could enhance treatment adherence, as reported in a related study carried out in Nigeria [27]. Support from friends and family can help patients feel less stressed psychologically and financially, especially in Ghana where the extended family structure is prevalent [28]. The findings also revealed that to restore confidence of patients in believing that going to the health facility can easily result in the exposure of their status, a secured lockers instead of the use of shelves in the clinics have been adopted. This was to ensure that patients records are intact as they will receive care and treatment for life.

This study also found that regular phone calls should be made to patients, especially those who are always getting to the time of refill. This implied that patients who don't have a record of their phone contact or who live in an area with inadequate telecommunications network would therefore fail to receive clinic appointment reminders and end up missing appointments. Clients who can be contacted need to be asked if they have experienced any side effects while taking the drugs. This will also offer the health staff another opportunity to counsel clients appropriately. This result corroborated a different study's finding that sending reminders through mobile message was linked to a higher chance of patients keeping their clinic appointments [29,30].

The practice of having a designated ART centres is discouraged in this current study, because it was noted to make clients feel more easily exposed to stigma and increases self-stigma among patients. This is because everybody knows that entering there means that the person is seeking this service. However, using an integrated care will offer the opportunity to other people too to walk in there freely with other conditions. In addition, it has been found in

this study that the involvement of religious leaders would serve as a retention measure of people on ARV to adhere to the drugs intake because people tend to believe in them and take their words more than the health workers. It was further revealed that if religious leaders can understand that HIV is not a spiritual disease, and educate their members, such clients would adhere to the treatment.

It was revealed from the current study that clients who were sick from other diseases and cannot get to the clinic for ARVs continue to receive and take their drugs to avoid the worsening of their condition. Additionally, this study's findings demonstrated that such clients were given the opportunity to get a trusted relative or individual from their community to come for the drugs on their behalf. The challenge in such circumstance, however, is the taking of sample for viral load testing from such clients. There is however no special support from either the district health directorate or other supporting organization (s) in such cases.

The use of some designated days for clients to come to the clinics for their ARVs was discouraged in this study area. This was practice to avoid the fear among clients that their status could easily be exposed. As a result, any day that client was due for the drugs, he or she could go to the centre for them. In most instances, clients were also advised not to wait for their drugs to get finished before they could go for refill. This was also to avoid situations where such clients may be unable to come on their exact refill day.

## Conclusion

In conclusion, this study highlights some of the major challenges faced in carrying out follow-up visits of patients on ART, including wrong addresses and phone numbers, poor telecommunication network, geographical location, poor documentation of patient information, and lack of means of transportation. However, the study also identifies several effective reengagement strategies, including supply of drugs through home visits, intensive education, engagement of CSOs, enhanced regular phone calls visits, and intensified education on HIV.

The findings call for an all-inclusive effort to be urgently made to reduce the prejudice and stigma at ART clinics. In this sense, healthcare workers' attitudes and behaviours toward clients who are HIV positive must be professional. This calls for frequent on-the-job training to raise awareness among health professionals on the need to refrain from stigma and discrimination. The results call for the Ghana AIDS Commission and Ghana Health Service to work with donor organizations to guarantee a steady supply of ART medicines at all ART centres to scale up and sustain the number of HIV positive individuals enrolled in ART.

## Supporting information

**S1 Checklist.**
(DOCX)

**S1 Data.**
(DOCX)

## Acknowledgments

We are grateful to the Ahafo Regional Health Directorate and Asunafo South District Health Directorate for granting us the permission to conduct this study. We also thank the facilities' In-charges and ART nurses for their assistants in getting the needed study respondents. To our research data collection assistants, our research team, and especially our participants involved, we say thank you.

## Author Contributions

**Conceptualization:** Robert Kogi.

**Data curation:** Robert Kogi, Emmanuel Asampong.

**Formal analysis:** Robert Kogi, Emmanuel Asampong.

**Funding acquisition:** Robert Kogi, Emmanuel Asampong.

**Investigation:** Robert Kogi, Theresa Krah.

**Methodology:** Robert Kogi, Theresa Krah, Emmanuel Asampong.

**Project administration:** Robert Kogi.

**Supervision:** Theresa Krah, Emmanuel Asampong.

**Validation:** Emmanuel Asampong.

**Visualization:** Emmanuel Asampong.

**Writing – original draft:** Robert Kogi, Theresa Krah, Emmanuel Asampong.

**Writing – review & editing:** Robert Kogi, Theresa Krah, Emmanuel Asampong.

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
