## [Decision Letter · Decision Letter 0]

6 Mar 2024

PONE-D-24-02076Factors influencing patients on antiretroviral therapy lost to follow up in Asunafo South District of Ahafo Region, GhanaPLOS ONE

Dear Dr. Kogi,

Thank you for submitting your manuscript to PLOS ONE. After careful consideration, we feel that it has merit but does not fully meet PLOS ONE’s publication criteria as it currently stands. Therefore, we invite you to submit a revised version of the manuscript that addresses the points raised during the review process.

**ACADEMIC EDITOR: **The manuscript is well written and the findings will be of immense importance toward improving retention on ART and VL suppression among people living with HIV in the study locality. To strengthen the manuscript, please respond to the following comments in addition to reviewers comments below.

Data availability: Please indicate that data is submitted under "additional documents"

Material and methods – please define “retention” and “LTFU" in the context of this study, using reliable references for the definitions.

Sampling method and sample size: Purposive sampling used - what criteria used in the selection of the participants (both patients and healthcare providers)

Data analysis – please indicate what coding approach was used (deductive vs inductive).

Discussion – line 555: what does “Major challenges in carrying out lost to follow up” mean? Please adjust the phrase.

Kind regards,

Ibrahim Jahun, MD, MSC, PhD

Academic Editor

PLOS ONE

3. In this instance it seems there may be acceptable restrictions in place that prevent the public sharing of your minimal data. However, in line with our goal of ensuring long-term data availability to all interested researchers, PLOS’ Data Policy states that authors cannot be the sole named individuals responsible for ensuring data access (http://journals.plos.org/plosone/s/data-availability#loc-acceptable-data-sharing-methods).

Reviewers' comments:

Reviewer's Responses to Questions

**Comments to the Author**

1. Is the manuscript technically sound, and do the data support the conclusions?

Reviewer #1: Yes

Reviewer #2: Partly

2. Has the statistical analysis been performed appropriately and rigorously? 

Reviewer #1: N/A

Reviewer #2: Yes

3. Have the authors made all data underlying the findings in their manuscript fully available?

Reviewer #1: Yes

Reviewer #2: Yes

4. Is the manuscript presented in an intelligible fashion and written in standard English?

Reviewer #1: Yes

Reviewer #2: Yes

5. Review Comments to the Author

Reviewer #1: There may be concerns about the novelty of the findings, but I believe that, as far as the Ahafo region of Ghana is concerned, this study speaks to challenges encountered by the HIV treatment and care system in the execution of its responsibilities. The methodology is sound and simple enough to address the issues identified.

I do have some concerns about language (grammar, etc.), but these are minor and revisions have been recommended.

Reviewer #2: This is a very interesting paper.

The authors should consider the following recommendations:

First, the study participants is a mix of healthcare workers and patients on antiretroviral therapy (ART). These two groups have different perspective on factors that influence loss to follow up. Also, considering the non-probability sampling technique that has a higher representation of healthcare workers. The paper will be better focussed if the perspectives of one group either the healthcare workers or patients on ART is presented in this qualitative research.

Secondly, the title should be modified to reflect the study participants see suggestion: "Factors influencing patients on antiretroviral therapy loss to follow up: a qualitative analysis of healthcare workers perspective."

6. PLOS authors have the option to publish the peer review history of their article (what does this mean?). If published, this will include your full peer review and any attached files.

Reviewer #1: No

Reviewer #2: **Yes: **Matthias Alagi

---

## [Author Response · Author response to Decision Letter 0]

13 Mar 2024

Academic Editor’s comments

Data availability: Please indicate that data is submitted under "additional documents: I have added that “the data is submitted under additional documents” in the submission system

Material and methods – please define “retention” and “LTFU" in the context of this study, using reliable references for the definitions: I have defined retention on care as “patients who, in the 12 months preceding this study, made at least 4 appointments, with at least 1 visit per quarter (14), and patients who are alive and receiving highly active antiretroviral therapy by the time of conducting this study (15).” And LTFU as “patient(s) not being on ART for more than one month (16)” with their appropriate references in page 6.

Sampling method and sample size: Purposive sampling used - what criteria used in the selection of the participants (both patients and healthcare providers): I have stated the criterion for selecting the respondents (healthcare providers) as “purposive sampling was used to identify the health workers based on their knowledge and role in the management of HIV patients” in page 6. However, that of the patients was left out because of the exclusion of the patients from the entire work as suggested by Reviewer #2

Data analysis – please indicate what coding approach was used (deductive vs inductive): I have stated that “We used both deductive and inductive coding approaches in this study. Deductive coding was done with predefined codes or categories being applied based on existing theory or literature (in the basic and organizing themes), while elements of inductive coding was done with themes and patterns allowed to emerge directly from the data (labelled codes) in Table 1.” In page 7

Discussion – line 555: what does “Major challenges in carrying out lost to follow up” mean? Please adjust the phrase: This was addressed as “Challenges facing conduct of patient loss to follow-up exercise”

We note that the grant information you provided in the ‘Funding Information’ and ‘Financial Disclosure’ sections do not match: The Financial Disclosure has been revised and added to the Cover Letter as "Funding for this project was provided by Access and Delivery Partnership (ADP), supported by the Government of Japan and led by the United Nations Development Programme, in collaboration with the World Health Organization’s Special Programme for Research and Training in Tropical Diseases (TDR) and PATH (Grant number: UGSPH/CIRT-M/003)."

Before we proceed with your manuscript, please also provide non-author contact information (phone/email/hyperlink) for a data access committee, ethics committee, or other institutional body to which data requests may be sent. Data for this study is attached as an additional file to supporting information in this manuscript submission.

Reviewer #1

I do have some concerns about language (grammar, etc.), but these are minor and revisions have been recommended. The manuscript has been revised entirely to correct minor identified grammar as indicated throughout the content

Reviewer #2

First, the study participants is a mix of healthcare workers and patients on antiretroviral therapy (ART). These two groups have different perspective on factors that influence loss to follow up. Also, considering the non-probability sampling technique that has a higher representation of healthcare workers. The paper will be better focussed if the perspectives of one group either the healthcare workers or patients on ART is presented in this qualitative research.

Secondly, the title should be modified to reflect the study participants see suggestion: "Factors influencing patients on antiretroviral therapy loss to follow up: a qualitative analysis of healthcare workers perspective." The content of the manuscript has been appropriately revised to focus on healthcare providers only. 

The title has also been modified as follows: “Factors influencing patients on antiretroviral therapy loss to follow up: A qualitative analysis of healthcare workers perspective”

---

## [Decision Letter · Decision Letter 1]

29 Apr 2024

PONE-D-24-02076R1Factors influencing patients on antiretroviral therapy loss to follow up: A qualitative analysis of healthcare workers perspectivePLOS ONE

Dear Dr. Kogi,

Thank you for submitting your manuscript to PLOS ONE and for addressing most of the comments shared with you during initial review. After careful consideration, we feel that there are few areas that require your attention as indicated by reviewer #2. Therefore, we invite you to submit a revised version of the manuscript that addresses the points raised by reviewer 2. You may also notice that we invited additional reviewer (expert in qualitative methodology) to review the methods section, and the reviewer has cleared the methods section.

Kind regards,

Ibrahim Jahun, MD, MSC, PhD

Academic Editor

PLOS ONE

Journal Requirements:

Reviewers' comments:

Reviewer's Responses to Questions

**Comments to the Author**

1. If the authors have adequately addressed your comments raised in a previous round of review and you feel that this manuscript is now acceptable for publication, you may indicate that here to bypass the “Comments to the Author” section, enter your conflict of interest statement in the “Confidential to Editor” section, and submit your "Accept" recommendation.

Reviewer #1: All comments have been addressed

Reviewer #2: All comments have been addressed

Reviewer #3: All comments have been addressed

2. Is the manuscript technically sound, and do the data support the conclusions?

Reviewer #1: Yes

Reviewer #2: Yes

Reviewer #3: Yes

3. Has the statistical analysis been performed appropriately and rigorously? 

Reviewer #1: Yes

Reviewer #2: N/A

Reviewer #3: Yes

4. Have the authors made all data underlying the findings in their manuscript fully available?

Reviewer #1: Yes

Reviewer #2: Yes

Reviewer #3: Yes

5. Is the manuscript presented in an intelligible fashion and written in standard English?

Reviewer #1: Yes

Reviewer #2: Yes

Reviewer #3: Yes

6. Review Comments to the Author

Reviewer #1: This paper is now ready to be published. It is interesting work and I look forward to seeing how it is platformed to improve HIV services in the area.

Reviewer #2: Thank you for addressing the earlier review comments. However, the following minor revisions are required:

1. Sampling method and sample size are not described under the methods and materials section. Kindly revise with a brief description of the sampling method and sample size of participants interviewed.

2. Lines 107 - 109 reads "Eligible HIV patients on ART, as well as those on ART but lost to follow-up, were contacted via phone calls by ART nurses to explain the study, and those who accepted were scheduled for face-to-face structured interviews." Kindly delete or explain why patients are being interviewed.

3. Most of lines 105 - 126 describes the study procedure. Kindly consider "Study Procedure" as a more appropriate subheading.

4. The result section should present sociodemographic and occupational characteristics of the healthcare workers interviewed as participants in this study.

Thank you once more for paying attention to details.

Reviewer #3: (No Response)

7. PLOS authors have the option to publish the peer review history of their article (what does this mean?). If published, this will include your full peer review and any attached files.

Reviewer #1: No

Reviewer #2: **Yes: **Alagi, Matthias

Reviewer #3: No

---

## [Author Response · Author response to Decision Letter 1]

8 May 2024

I would like to bring to your attention that I have responded to all the comments made on our manuscript based on reviewer #2 comments. The responses to the individual comments are appropriately reported in the table below for your consideration.

Thank you.

#1. Sampling method and sample size are not described under the methods and materials section. Kindly revise with a brief description of the sampling method and sample size of participants interviewed.

Response: This section of the manuscript has been revised as follows: “The total sample used in this study was seven (7). This included one doctor, one physician assistant, three ART clinic staff, two community health workers (who were experienced in seeking patients in the community). 

Purposive sampling was used to identify the health workers based on their knowledge and role in management of HIV patients.”

#2. Lines 107 - 109 reads "Eligible HIV patients on ART, as well as those on ART but lost to follow-up, were contacted via phone calls by ART nurses to explain the study, and those who accepted were scheduled for face-to-face structured interviews." Kindly delete or explain why patients are being interviewed.

Response: The sentence "Eligible HIV patients on ART, as well as those on ART but lost to follow-up, were contacted via phone calls by ART nurses to explain the study, and those who accepted were scheduled for face-to-face structured interviews." Has been deleted from Lines 107-109.

#3. Most of lines 105 - 126 describes the study procedure. Kindly consider "Study Procedure" as a more appropriate subheading.

Response: A subheading in Line 110 has been added as "Study Procedure"

#4. The result section should present sociodemographic and occupational characteristics of the healthcare workers interviewed as participants in this study.

Response: Sociodemographic and occupational characteristics of the healthcare workers have been added in Lines 145-150

---

## [Decision Letter · Decision Letter 2]

15 May 2024

Factors influencing patients on antiretroviral therapy loss to follow up: A qualitative analysis of healthcare workers perspective

PONE-D-24-02076R2

Dear Dr. Kogi,

We’re pleased to inform you that your manuscript has been judged scientifically suitable for publication and will be formally accepted for publication once it meets all outstanding technical requirements.

Kind regards,

Ibrahim Jahun, MD, MSC, PhD

Academic Editor

PLOS ONE

Additional Editor Comments (optional):

Reviewers' comments:

Reviewer's Responses to Questions

**Comments to the Author**

1. If the authors have adequately addressed your comments raised in a previous round of review and you feel that this manuscript is now acceptable for publication, you may indicate that here to bypass the “Comments to the Author” section, enter your conflict of interest statement in the “Confidential to Editor” section, and submit your "Accept" recommendation.

Reviewer #2: All comments have been addressed

2. Is the manuscript technically sound, and do the data support the conclusions?

Reviewer #2: Yes

3. Has the statistical analysis been performed appropriately and rigorously? 

Reviewer #2: Yes

4. Have the authors made all data underlying the findings in their manuscript fully available?

Reviewer #2: Yes

5. Is the manuscript presented in an intelligible fashion and written in standard English?

Reviewer #2: Yes

6. Review Comments to the Author

Reviewer #2: (No Response)

7. PLOS authors have the option to publish the peer review history of their article (what does this mean?). If published, this will include your full peer review and any attached files.

Reviewer #2: **Yes: **Matthias Alagi

---

## [Editor Report · Acceptance letter]

21 May 2024

PONE-D-24-02076R2 

PLOS ONE

Dear Dr. Kogi, 

I'm pleased to inform you that your manuscript has been deemed suitable for publication in PLOS ONE. Congratulations! Your manuscript is now being handed over to our production team.

Kind regards, 

on behalf of

Dr. Ibrahim Jahun 

Academic Editor

PLOS ONE